Using lidar to assess the development of structural diversity in forests undergoing passive rewilding in temperate Northern Europe

http://orcid.org/0000-0001-9166-1418 Thers Henrik 1 2 thers@agro.au.dk
http://orcid.org/0000-0002-3045-4607 Bøcher Peder Klith 2 3
http://orcid.org/0000-0002-3415-0862 Svenning Jens-Christian 2 3
1 Department of Agroecology, Aarhus University , Tjele , Denmark
2 Center for Biodiversity Dynamics in a Changing World (BIOCHANGE), Department of Bioscience, Aarhus University , Aarhus , Denmark
3 Section for Ecoinformatics and Biodiversity, Department of Bioscience, Aarhus University , Aarhus , Denmark
Huang Cho-ying
Electronic publication date: 2019 Jan 14
Publication date: 2019
Volume: 6
Electronic Location ID: e6219
Received 2018 Jun 4; Accepted 2018 Dec 5
Copyright: © 2019 Thers et al.
Copyright year: 2019
Copyright holder: Thers et al.
License: This is an open access article distributed under the terms of the Creative Commons Attribution License, which permits unrestricted use, distribution, reproduction and adaptation in any medium and for any purpose provided that it is properly attributed. For attribution, the original author(s), title, publication source (PeerJ) and either DOI or URL of the article must be cited.
License URL: https://creativecommons.org/licenses/by/4.0/

Keywords: Natural regeneration, Lidar, Passive rewilding, Disturbances, Vegetation structure

Funding: VILLUM FONDEN 16549 through Jens-Christian Svenning’s VILLUM Investigator project “Biodiversity Dynamics in a Changing World” This work was supported by VILLUM FONDEN (grant 16549) through Jens-Christian Svenning’s VILLUM Investigator project “Biodiversity Dynamics in a Changing World.” The funders had no role in study design, data collection and analysis, decision to publish, or preparation of the manuscript.

==============================
Forested areas are increasing across Europe, driven by both reforestation programs and farmland abandonment. While tree planting remains the standard reforestation strategy, there is increased interest in spontaneous regeneration as a cost-effective method with equal or potentially greater benefits. Furthermore, expanding areas of already established forests are left for passive rewilding to promote biodiversity conservation. Effective and objective methods are needed for monitoring and analyzing the development of forest structure under these management scenarios, with airborne laser scanning (lidar: light detection and ranging) being a promising methodology. Here, we assess the structural characteristics and development of unmanaged forests and 28- to 78-year old spontaneously regenerated forests on former agricultural land, relative to managed forests of similar age in Denmark, using 25 lidar-derived metrics in 10- and 30-m grid cells. We analyzed the lidar-derived cell values in a principal component analysis (PCA) and interpreted the axes ecologically, in conjunction with pairwise tests of median and variance of PCA-values for each forest. Spontaneously regenerated forest in general had increased structural heterogeneity compared to planted and managed forests. Furthermore, structural heterogeneity kept increasing in spontaneously regenerated forest across the maximal 78-year timespan investigated. Natural disturbances showed strong impacts on vegetation structure, leading to both structural homogeneity and heterogeneity. The results illustrate the utility of passive rewilding for generating structurally heterogeneous forested nature areas, and the utility of lidar surveys for monitoring and interpreting structural development of such forests.

Introduction

Forest structure is important for biodiversity, notably via providing heterogeneous environments in terms of available resources and structural composition (Simonson, Allen & Coomes, 2014; Stein, Gerstner & Kreft, 2014; Tews et al., 2004). Commercially and intensively managed forests often show less structural heterogeneity compared to unmanaged forests due to establishment strategy (planted as even-aged monocultures or naturally regenerated as fenced stands without gaps), and due to prevention of natural dynamics (harvesting the biologically young trees, by drainage, thinning and pest control) (Christensen & Emborg, 1996; Nielsen & Jensen, 2007; Richnau, Wistrom & Nielsen, 2012). The homogeneous structure of deciduous temperate forests under traditional silvicultural management has put a wide range of forest-associated species at risk, affecting both fungi, animals and plants (Christensen & Emborg, 1996; Hofmeister, Hosek & Brabec, 2015; Ishii, Tanabe & Hiura, 2004). Hence, one relevant aim for forest restoration is to facilitate the development of more heterogeneous forest structure than seen in managed forests (Götmark, 2013). Passive rewilding, defined as spontaneous ecological dynamics without management (Svenning et al., 2016), is one restoration strategy that is thought to lead to more heterogeneous forests compared to managed forests. Pathways toward forests under passive rewilding can be: (1) ceasing the silvicultural management of existing forests and (2) natural regeneration of new forest areas, for example, on abandoned farmland (Schnitzler, 2014).

Forest cover in Europe is increasing due to active reforestation as well as abandonment of farmland (Navarro & Pereira, 2012; Verheyen, Vanhellemont & Auge, 2016). In Europe, between 10 and 29 million ha of agricultural land is predicted to become abandoned between 2000 and 2030 (Verburg & Overmars, 2009), much of it becoming available for active reforestation or passive rewilding. Advantages and drawbacks of various ways of forest establishment have been investigated in terms of costs, biodiversity, public opinion, carbon sequestration and other ecosystem services, and natural regeneration is considered the cheapest and slowest (Benayas, Bullock & Newton, 2008; Cramer, Hobbs & Standish, 2008; Nielsen & Jensen, 2007; Prach & Pyšek, 2001; Verheyen, Vanhellemont & Auge, 2016). However, natural regeneration is also the most uncertain method, with an outcome that is hard to predict (Benayas, Bullock & Newton, 2008; Bullock, Moy & Pywell, 2002; Prach & Pyšek, 2001; Schnitzler, 2014). Among factors influencing the development of vegetation structure following natural regeneration are seed supply, seed predation, herbivores, abiotic site conditions (e.g., pH and soil moisture), competition from herbs, non-indigenous plant species, disturbance regime and former land use; factors which interact in complex ways (Hobbs & Cramer, 2007a; Nilsson, Hedin & Niklasson, 2001; Prach & Pyšek, 2001). In addition, the speed of secondary succession varies. Long delays in succession can occur due to arresting factors such as high water level, drought and competition from herbs and grasses (Prach & Pyšek, 1994, 2001). Such prolonged vegetation development secures a mixture of open areas and denser vegetation, which has been found to increase the overall forest biodiversity (Sebek et al., 2015). Consequently, the various outcomes of complex natural regeneration make it difficult to optimize designs in targeting the specific purpose of reforestation (Navarro & Pereira, 2012).

Existing forests that are left for passive rewilding are more likely to be affected by natural disturbances due to windfall, insects and moist conditions and have more and larger dead trunks compared to managed forests (Nilsson, Hedin & Niklasson, 2001). The level and type of disturbance are decisive to the development of forest structure (Syrjänen et al., 1994), and an intermediate disturbance level is believed to provide the most heterogeneous and species-rich habitat (Roberts & Gilliam, 1995). It follows that forests allowed to undergo rewilding can be expected to increase heterogeneity and thereby create niches available to a more diverse species composition.

In order to be able to advise managers on the effectiveness of passive rewilding in producing structurally heterogeneous forests of high value for biodiversity, along with optimized management actions in the rewilding process, it is necessary to monitor and evaluate the structural development of vegetation left for passive rewilding. Lidar (light detection and ranging) data has proven to be efficient for measuring forest structure on a scale relevant for biodiversity (Bässler et al., 2011; Thers et al., 2017; Zellweger et al., 2016), and in recent years further studies have presented lidar-based methods for differentiating vegetation structure in stands of varying management and site history. Valbuena et al. (2016) assessed recognition of forest structural type by ALS-based gini coefficients and related the findings to management in Finnish boreal stands. Ehbrecht et al. (2016) and Listopad et al. (2018) measured effective numbers of forest layers in Germany and shrub regeneration in Portugal, respectively, using terrestrial laser scanning, and proved significant correlations to traditional forestry metrics. Listopad et al. (2018) also compared the findings to the site grazing history. However, there is a lack of papers focusing on the use of airborne laser scanning (ALS)-derived information to add additional information to the specific site history and to interpret the findings in relation to ecological theory, hereby strengthening the foundation for rewilding projects.

We aim to use simple wall-to-wall ALS point cloud-derived metrics treated by principal component analysis (PCA) to analyze a unique Danish rewilding site (the 64-ha island of Vorsø) and to relate the findings to the existing ecological theory of natural disturbances and spontaneous vegetation development. Hereby, merging a remote sensing survey and rewilding ecology relevant to policy makers and managers. We do this by comparing rewilded stands to managed stands of similar age, soil and climatic conditions. The Vorsø case provides the possibility to evaluate two different starting points for passive rewilding—spontaneous regeneration on abandoned farmland and the ceasing of management actions in existing forests.

Methods

Study sites

Four localities where chosen for the examination and comparison of forest structure by lidar (Figs. 1 and 2). The sites are situated within a 900-km2 area delimited by the UTM zone 32N coordinates: Northering: 6190000–6220000 and Eastering: 550000–580000. The main locality was Vorsø, an island in Horsens Fjord in eastern Jutland (Denmark). Vorsø was chosen because of the unique and well-documented vegetation history (Halberg & Gregersen, 2010). We divided Vorsø into seven areas, hereafter named zones, according to the history of the island. The zones are named with the letter V for Vorsø, followed by Y (young), M (mid-aged) or O (old) corresponding to the site history (vegetation age) of the zone. The numbers (1, 2,..) differentiates zones and areas of similar age. Areas influenced by humans, such as buildings, roads, and old gardens, were excluded from the seven zones of Vorsø.

Figure 1 Site locations.

(A) The geographical position of the four sites (Three managed forests and Vorsø) investigated in this paper marked with black stars in the eastern part of Jutland (Denmark). (B) The position of Denmark in Europe (Marked with light edges). (C) The island of Vorsø; V_O1 (6.11 ha); V_O2 (2.65 ha); V_O3 (0.38 ha) (labeled below the zone); V_M1 (15.06 ha); V_M2 (9.09 ha); V_Y2 (5.09 ha); V_Y1 (8.89 ha). (D) The extent of the Tran_Y3 oak (Quercus Robur) forest (4.51 ha). (E) Fløj_Y4, the extent of the mixed maple (Acer pseudoplatanus) and beech (Fagus sylvatica) forest (0.92 ha). (F) Sten_M/O, the extent of the predominantly beech (Fagus Sylvatica) forest (20.42 ha). A minor stand of spruce (Picea abies) is recognizable as a dark area in the lower part of the forest. See Fig. 2 for photos of the stands. DDO@Land 2016, © COWI A/S.

Figure 2 Photos of the managed stands and selected zones of Vorsø.

(A) A representative site of the managed oak forest of Tran_Y3. (B) A representative site of the managed maple and beech forest of Fløj_Y4. (C) A representative site of the predominated beech forest of Sten_M/O. (D) An example of the homogeneous structure in the sycamore maple (Acer pseudoplatanus) -dominated eastern and southern part of V_M2. Note the structural similarity to the Sten_M/O forest (C). (E) The succession of the otherwise old V_O2 has started over, due to the impact of cormorants (Phalacrocorax carbo), and consists of scattered young trees and shrubs, mainly elderberry (Sambucus nigra). (F) An example of the mixed structure of scrub, gaps, large trees and dead wood only found in the V_M1.

A foundation bought and protected the island in 1929; leaving the three existing forest patches (V_O1, V_O2 and V_O3) and the main part of the farmland (V_M1 and V_M2) for passive rewilding. The passive rewilding process of the three forests were initiated by the destruction of the drainage and the eradication of bigger stands of non-indigenous tree species. In 1979 farming ceased in the remaining farmland (V_Y1 and V_Y2). No management, planting or seeding actions have been performed during the years of protection, although cormorants (Phalacrocorax carbo) was regulated before 1979 to keep numbers down. The main influencing factors since protection have been tree-killing excrements from the cormorant colonies (heavy impact on V_O2, and minor to medium impact on V_O1 and V_M1), windfall (mainly V_O1), roe deer (Capreolus capreolus), Dutch elm disease, along with heavy seed supply of sycamore maple to V_M2 from V_O2 and V_O3.

At the time of our survey the dominant tree species were oak (Quercus robur), maple (Acer pseudoplatanus), ash (Fraxinus excelsior), beech (Fagus sylvatica) and elm (Ulmus glabra, heavily reduced by Dutch elm disease), as well as alder (Alnus glutinosa) in wet areas. Frequent occurring shrub species were elderberry (Sambucus nigra), blackthorn (Prunus spinosa), raspberry (Rubus idaeus), single-seeded hawthorn (Crataegus monogyna), cherry plum (Prunus cerasifera), dog rose (Rosa canina) and willow species (Salix cinerea, Salix caprea).

Three managed forests nearby Vorsø representing typical management scenarios and hence relevant for comparison to the Vorsø passive rewilding case were pointed out: – Tranbjerg (hereafter Tran_Y3), a 4.51 ha young managed stand planted in 1983 on former farmland. The main tree species is oak (Quercus robur). Drainage is unchanged. There has been no fencing. Managers performed weeding after planting, and thinning of the stand has been carried out a couple of times.

– Fløjstrup (hereafter Fløj_Y4), a 0.92 ha young managed stand established in 1983 by natural regeneration after a clear-cut. The main tree species is sycamore maple (Acer pseudoplatanus) mixed with beech (Fagus sylvatica) and ash trees (Fraxinus excelsior). There is only surface drainage and the stand was too wet for forest machinery to operate. The stand was fenced during regeneration and thinning has been carried out 3–4 times to control numbers of trees and remove some tree species.

– Stensballe (hereafter Sten_M/O), a 20.42 ha semi-old stand of mainly broadleaved trees under commercial forest management. The main species is beech (Fagus sylvatica). Mature trees are selectively harvested and the upbringing of new trees is secured by natural regeneration, leading to a mixed stand of age classes.

Lidar-derived metrics

Lidar-data was acquired with an airborne Optech Altm Gemini Laser Altimeter system in March 2007 (kortforsyningen.dk, 2015) with a footprint of 40 cm. The point cloud density was one return per 2.2 m2 on bare ground and included up to two returns per pulse with a vertical and horizontal accuracy of 0.67 and 0.06 m, respectively. We derived 25 metrics from the height-normalized point cloud after deleting overlapping returns. The 25 metrics were found in existing lidar literature and selected on a criteria of being used in comparison to manual ground surveys in previous studies of for being alike such metrics (Table S1). We performed calculations and processing in the ArcGIS (version 10.2.2.3552; Esri ArcMap, Redlands, California, USA) software and LasTOOLS (Version 141117; Rapidlasso, GmbH, Gilching, Germany). Basic grid cells measured 10 × 10 m ensuring sufficient returns per cell for statistical treatment.

We furthermore computed 25 metrics of horizontal heterogeneity based on the first 25 metrics, by calculating the standard deviation of the values for each cell (10 ×10 m) and the eight surrounding cells and ascribing this SD-value to each cell. Thereafter the cells were aggregated into bigger cells of 30 × 30 m, by taking the mean of the SD-values of nine adjacent cells to reduce pseudo-replication by adjacent cells influencing each other in the SD calculation.

Statistical analysis

We extracted the values from each 10 × 10 m cell of the 25 metrics for vertical structure and each 30 × 30 m cell of the 25 metrics for horizontal heterogeneity and exported it to the R (R i386 3.1.2Ink; R Core Team, 2014) statistical software. We scaled and centered the values and completed one PCA (hereafter PCA_10m) based on values from the 10 × 10-m grid cells (n = 7,319), and another one (hereafter PCA_30m) based on the 30 × 30-m grid cells (n = 1,002).

Based on a threshold of about 90% explained variance, the first five principal axes were tested for significant differences in medians and variances among the three managed forests and seven rewilded zones of Vorsø focusing on the even-aged stands (Table 1), using Kruskal–Wallis as a non-parametric test for medians and Levene’s test for variance. In addition, we included extreme PCA axes values as a measure of unique vegetation structure in the analyses.

Table 1 Groupings for age-specific pairwise comparisons of rewilded forest zones on Vorsø and nearby managed forests.

Groupings	Zones of Vorsø and managed forests	
Young stands	V_Y1; V_Y2; Tran_Y3; Fløj_Y4	
Mid-aged stands	V_M1; V_M2; Sten_M/O	
Old stands	V_O1; V_O2; V_O3; Sten_M/O	
Note:

The two zones, V_Y1 and V_Y2, have similar histories and Tran_Y3 and Fløj_Y4 are approximately of the same age. V_M1 and V_M2 have similar histories. V_O1, V_O2 and V_O3 were rewilded simultaneously, but their initial conditions differed, for example, in terms of species composition. The managed Sten_M/O forest is included in two age groups because the age of the oldest trees corresponds roughly to the age of the mid-aged group, but at the same time the managed forest is persistently kept at this stage due to tree harvest. Sten_M/O therefore also serves as an example of the managed alternative to the rewilded forests in this study.

Results

Within the young stands, the Vorsø zones have the highest variances on four of five axes of PCA_10m compared to the young managed forests (Figs. 3 and 4; Table 2).

Figure 3 Boxplots of PCA values for PCA_10m and PCA_30m.

(A–E) Boxplots of the values of axes 1, 2, 3, 4 and 5 of the PCA_10m and (F–J) Boxplots of the values of axes 1, 2, 3, 4 and 5 of the PCA_30m. Number of 10-m cells in each area in parentheses and number of 30-m cells in brackets. V_O1 of Vorsø (n = 618) [n = 73], V_O2 of Vorsø (n = 263) [n = 31], V_O3 of Vorsø (n = 36) [n = 4], Sten_M/O (n = 2,041) [n = 275], V_M2 of Vorsø (n = 912) [n = 120], V_M1 of Vorsø (n = 1,496) [n = 199], V_Y1 of Vorsø (n = 511) [n = 74], V_Y2 of Vorsø (n = 899) [n = 127], (I), Tran_Y3 (n = 450) [n = 79] and (J), Fløj_Y4 (n = 93) [n = 20]. There is a rough gradient of age, placing the oldest stands to the left and the youngest to the right. The whiskers mark the minimum and maximum values omitting outliers. The 5 and 95 percentiles are marked with dots. Note that the relative variance alternates between the stands and axes, meaning that the gradients (axes) are varying, suited for differentiating specific types of structure.

Figure 4 Maps of PCA values.

(A) Values of PCA_10m axis 1, which primarily is a gradient of vegetation height. The highest values are found among the oldest vegetation of V_O1 and Sten_M/O, along with the younger stand V_M2, which where rapidly colonized by sycamore maple. Red-colored cells indicate widespread open areas in three of the Vorsø zones. (B) Values of PCA_10m axis 5, which is a gradient from bare ground to low and sparse vegetation, subscribing intermediate values to all kinds of taller vegetation. Low values (red) specify the bare ground. Axis 5 is one axis that differentiates the two youngest zones (V_Y1 and V_Y2) on both median and variance, though it is visible that the two former fields contain the same vegetation structural components according to the mapping of axis values. For both A and B, Vorsø zones corresponds to Fig. 1.

Table 2 Pairwise comparisons of stands within groupings according to age.

A		PCA_10m	PCA_30m	
		Variance (σ2)	V_Y1	V_Y2	Tran_Y3	Fløj_Y4	Variance (σ2)	V_Y1	V_Y2	Tran_Y3	Fløj_Y4	
V_Y1	dissim.		x	6	10	7		x	0	8	6	
V_Y1	PC1	5.91	x	***	***/+++	***/+++	7.55	x	non-sign.	***/+++	*	
V_Y1	PC2	4.32	x	***/+	***/+++	+++	3.15	x	non-sign.	***/+++	***/+	
V_Y1	PC3	0.90	x	non-sign.	***/+++	***	1.56	x	non-sign.	***/+++	***	
V_Y1	PC4	1.54	x	++	*/+++	***/+++	2.38	x	non-sign.	**/+++	+++	
V_Y1	PC5	2.33	x	*/+++	***/+++	+++	0.28	x	non-sign.	non-sign.	*	
V_Y2	dissim.		6	x	9	7		0	x	7	5	
V_Y2	PC1	6.08	***	x	***/+++	***/+++	6.88	non-sign.	x	***/+++	***/+	
V_Y2	PC2	3.31	***/+	x	***/+++	+++	2.88	non-sign.	x	***/+++	***/++	
V_Y2	PC3	0.69	non-sign.	x	***/+++	***	1.56	non-sign.	x	***/+++	**	
V_Y2	PC4	0.95	++	x	+++	***/+++	2.49	non-sign.	x	+++	non-sign.	
V_Y2	PC5	1.30	*/+++	x	*/+++	+++	0.20	non-sign.	x	non-sign.	non-sign.	
Tran_Y3	dissim.		10	9	x	8		8	7	x	3	
Tran_Y3	PC1	0.27	***/+++	***/+++	x	***/+++	1.00	***/+++	***/+++	x	***	
Tran_Y3	PC2	0.16	***/+++	***/+++	x	***	0.07	***/+++	***/+++	x	***	
Tran_Y3	PC3	0.24	***/+++	***/+++	x	+++	0.29	***/+++	***/+++	x	*	
Tran_Y3	PC4	0.05	*/+++	+++	x	***/+++	0.09	**/+++	+++	x	non-sign.	
Tran_Y3	PC5	0.04	***/+++	*/+++	x	***/+++	0.19	non-sign.	non-sign.	x	non-sign.	
Fløj_Y4	dissim.		7	7	8	x		6	5	3	x	
Fløj_Y4	PC1	0.66	***/+++	***/+++	***/+++	x	0.65	*	***/+	***	x	
Fløj_Y4	PC2	0.22	+++	+++	***	x	0.23	***/+	***/++	***	x	
Fløj_Y4	PC3	0.92	***	***	+++	x	0.74	***	**	*	x	
Fløj_Y4	PC4	0.17	***/+++	***/+++	***/+++	x	0.16	+++	non-sign.	non-sign.	x	
Fløj_Y4	PC5	0.16	+++	+++	***/+++	x	0.35	*	non-sign.	non-sign.	x	
B		PCA_10m	PCA_30m	
		Variance (σ2)	V_M2	V_M1	Sten_M/O		Variance (σ2)	V_M2	V_M1	Sten_M/O		
V_M2	dissim.		x	9	9			x	4	7		
V_M2	PC1	7.48	x	***/+	***/+++		8.65	x	non-sign.	***		
V_M2	PC2	0.85	x	***/+++	***		3.93	x	***	***/+++		
V_M2	PC3	1.46	x	***/+++	***/+++		1.07	x	***/++	***		
V_M2	PC4	0.55	x	***/+++	***/+++		0.98	x	***	***/+		
V_M2	PC5	0.30	x	+++	***/+++		0.91	x	non-sign.	***		
V_M1	dissim.		9	x	10			4	x	10		
V_M1	PC1	6.32	***/+	x	***/+++		6.09	non-sign.	x	***/+		
V_M1	PC2	3.12	***/+++	x	***/+++		3.57	***	x	***/+++		
V_M1	PC3	0.66	***/+++	x	***/+++		2.58	***/++	x	***/+++		
V_M1	PC4	0.79	***/+++	x	***/+++		1.68	***	x	***/+++		
V_M1	PC5	1.28	+++	x	***/+++		1.68	non-sign.	x	***/+		
Sten_M/O	dissim.		9	10	x			7	10	x		
Sten_M/O	PC1	3.59	***/+++	***/+++	x		3.52	***	***/+	x		
Sten_M/O	PC2	1.06	***	***/+++	x		1.03	***/+++	***/+++	x		
Sten_M/O	PC3	2.42	***/+++	***/+++	x		1.39	***	***/+++	x		
Sten_M/O	PC4	1.99	***/+++	***/+++	x		0.44	***/+	***/+++	x		
Sten_M/O	PC5	0.16	***/+++	***/+++	x		0.80	***	***/+	x		
C		PCA_10m	PCA_30m	
		Variance (σ2)	V_O1	V_O2	V_O3	Sten_M/O	Variance (σ2)	V_O1	V_O2	V_O3	Sten_M/O	
V_O1	dissim.		x	7	4	8		x	4	0	7	
V_O1	PC1	6.58	x	***/+++	+	***/+++	21.63	x	***	non-sign.	***/+++	
V_O1	PC2	0.57	x	***	non-sign.	non-sign.	1.55	x	***	non-sign.	*	
V_O1	PC3	1.66	x	***/+++	***	***/+++	1.85	x	non-sign.	non-sign.	***	
V_O1	PC4	0.76	x	+++	*	*/+++	0.41	x	***	non-sign.	***	
V_O1	PC5	0.24	x	***	***	***/+	2.78	x	***	non-sign.	***/+++	
V_O2	dissim.		7	x	7	7		4	x	0	2	
V_O2	PC1	2.07	***/+++	x	***	***/+	8.94	***	x	non-sign.	non-sign.	
V_O2	PC2	0.66	***	x	***	***	0.82	***	x	non-sign.	***	
V_O2	PC3	0.47	***/+++	x	***/+++	***/+++	0.58	non-sign.	x	non-sign.	***	
V_O2	PC4	0.10	+++	x	***/+	+++	0.28	***	x	non-sign.	non-sign.	
V_O2	PC5	0.21	***	x	***	***	0.55	***	x	non-sign.	non-sign.	
V_O3	dissim.		4	7	x	2		0	0	x	0	
V_O3	PC1	1.60	+	***	x	non-sign.	0.56	non-sign.	non-sign.	x	non-sign.	
V_O3	PC2	0.24	non-sign.	***	x	non-sign.	0.39	non-sign.	non-sign.	x	non-sign.	
V_O3	PC3	2.81	***	***/+++	x	non-sign.	0.37	non-sign.	non-sign.	x	non-sign.	
V_O3	PC4	0.35	*	***/+	x	***	0.32	non-sign.	non-sign.	x	non-sign.	
V_O3	PC5	0.16	***	***	x	***	0.42	non-sign.	non-sign.	x	non-sign.	
Sten_M/O	dissim.		8	7	2	x		7	2	0	x	
Sten_M/O	PC1	3.59	***/+++	***/+	non-sign.	x	3.52	***/+++	non-sign.	non-sign.	x	
Sten_M/O	PC2	1.06	non-sign.	***	non-sign.	x	1.03	*	***	non-sign.	x	
Sten_M/O	PC3	2.42	***/+++	***/+++	non-sign.	x	1.39	***	***	non-sign.	x	
Sten_M/O	PC4	1.99	*/+++	+++	***	x	0.44	***	non-sign.	non-sign.	x	
Sten_M/O	PC5	0.16	***/+	***	***	x	0.80	***/+++	non-sign.	non-sign.	x	
Note:

Pairwise tests of medians and variances of the first five axes of the PCA_10m (left part of tables) and PCA_30m (right part of tables). Kruskal–Wallis and Levene’s test was used to test for differences of medians and variances, respectively. Asterisks (*) indicate significance for medians, and plusses (+) indicate significance for variance. Non-significance of both medians and variance is indicated with non-sign. Significance levels have been Bonferroni-corrected by dividing 0.05 with 225. The corrected level of significance used here was 0.000222. */+ = below 0.000222, **/++ = below 0.00001, ***/+++ = below 0.000001. The columns of variance (σ2) list variance values, and the largest variance values for the axes within each grouping are in bold and italics. The rows marked dissim. list, how many of the 10 pairwise tests that are significantly different as a measure of dissimilarity between two areas. X indicates non-relevant comparison. A: Pairwise tests between younger stands (V_Y1, V_Y2, Tran_Y3 and Fløj_Y4). B: Pairwise tests between mid-aged stands (V_M1, V_M2 and Sten_M/O). C: Pairwise tests between older stands (V_O1, V_O2, V_O3 and Sten_M/O). A complete table of all pairwise comparisons without regarding groupings are present in Tables S2 and S3 for PCA_10m and PCA_30m, respectively.

The PCA_30m supports those findings, showing significantly larger variance on four of the five axes and a significantly smaller median on axis 1 compared to both managed stands (Table 2).

The analysis points out the similarity between the two youngest rewilded zones of Vorsø, despite significantly different variances and medians of axes 2 and 5 (PCA_10m), visual analyzing of the map output clarifies overall similarity of the two youngest Vorsø zones (Fig. 4; Figs. S1 and S2). The PCA_30m confirms this, as no significant differences are found between V_Y1 and V_Y2 (Table 2). Furthermore, low values in PCA_10m axis 2 show, in compliance with reality, that gaps are present in V_Y1 and V_Y2 (and V_M1) while not in Tran_Y3 and Fløj_Y4.

In contrast to the similar development of the two youngest rewilded farmlands, the two mid-aged zones of Vorsø, V_M1 and V_M2, are significantly different in nine of 10 pairwise comparisons of PCA_10m axes values, which correspond to the number of significant differences to the managed Sten_M/O forest. The southern and eastern parts of V_M2 mimics the Sten_M/O, which is not the case for the V_M1 (visible at Fig. 4; Fig. S2). According to the tests of the PCA_30m, the two mid-aged rewilded zones are found less different with a dissimilarity value of 4, compared to 7 and 10 when compared to Sten_M/O (Table 2). Furthermore, extremely low values of PCA_30m axis 3 clarify that V_M1 keeps evolving new types of heterogeneity compared to the 50-year younger vegetation of V_Y1 and V_Y2. An example of what the two latter cannot produce, is the mix of older trees, scrub, gaps and dead wood (Fig. 2F; Fig. S2). In addition, after 78 years of spontaneous regeneration, the V_M1 still holds gaps surrounded by shrubs, which is detected by PCA_10m axes 2 and 5 (low values) and PCA_30m axis 2 (high values) (Figs. 3B, 3D and 3G, see Table S4 for axes interpretation).

Comparisons within the oldest group of stands show that they all differ according to vegetation structure, and that V_O3 presumably contains too few cells for statistical analyses, especially in PCA_30m, where no significant differences are found at all. Sten_M/O and V_O1 have the highest dissimilarity values on both PCA_10m and PCA_30m; the latter one pointing at V_O1 as the most heterogeneous by having significantly larger variances on axes 1 and 5, besides extreme low values and a smaller median on axis 1, meaning more heterogeneous vegetation according to height (Fig. 3; Fig. S2).

The PCA axes were interpretable to some extent by analyzing the loadings of metrics to the eigenvectors in combination with ground inspections, although; especially the PCA_30m results were complex due to the underlying calculations (Tables S4–S6). The PCA_30m seemed to be especially suitable to detect edge vegetation along glades and in addition, extreme cell values of the PCA_30m axes were useful to identify unique vegetation structural attributes, for example, in the V_M1 and V_O1 (Fig. S2).

Discussion

Rewilding of abandoned farmland

The youngest zones on Vorsø and the youngest managed forests (Tran_Y3 and Fløj_Y4) have developed completely differently, which we conclude must be due to the different afforestation techniques used, as well as the related management (fencing and weeding in Tran_Y3 and Fløj_Y4). The two latter are evolving homogeneously in line with previous investigations of intensively managed forests (Fuller, Foster & McLachlan, 1998; Nilsson, Hedin & Niklasson, 2001). The larger median values for the managed stands on PCA_10m axis 1 clearly indicate that the dense establishment led to a higher growth rate, even compared to V_M1, which has had 50 more years to grow tall vegetation. A recent study modelling post-disturbance spontaneous regeneration in the Bavarian Forest National Park, Germany, showed uniform development of mean tree height across five sites over 80 years, despite differences in their starting points (Hill et al., 2017). In contrast, the stand heights investigated in this paper develop differently. Since soil and climatic conditions are similar within the study area, we assign those structural differences to the method of establishment.

The relatively similar vegetation development in the two young Vorsø zones (V_Y1 and V_Y2) indicates predictability of underlying controlling ecological factors, also found in other studies (Bullock, Moy & Pywell, 2002; Prach & Pyšek, 2001), despite the complex interactions between the factors influencing development. The similar development suggests that spontaneous regeneration might be predictable, when conditions are well known as assumed by ongoing vegetation development modelling, for example, (Hill et al., 2017).

The two fields abandoned in 1929 (V_M1 and V_M2) also predominantly shared the same conditions. However, they developed quite distinct vegetation structure, identified by significantly different median and variance values in nine of 10 cases (PCA_10m) (Table 2). Different seed input is concluded as being the main reason for this divergence (Halberg & Gregersen, 2010). The prevailing western wind provided an effective vector for seeds of the fast-growing Acer pseudoplatanus trees from V_O2 and V_O3 to V_M2, causing the divergence between V_M2 and V_M2, which underlines the broadly acknowledged importance of seed banks and seed supplies (Benayas, Bullock & Newton, 2008; De Steven, 1991; Olsson, 1987). The ongoing diverging development of vegetation through 78 years in the two abandoned field sites was hence largely predictable when general successional ecology was considered and are in line with a recent study concluding that old-growth spatial pattern is determined by the initial patterns of regeneration (Hill et al., 2017). Thus, the combination of the recorded site history and the statistical result presented in the two above pairwise comparisons, despite different outcomes, underpin that designing spontaneous regeneration projects with specific aims for the vegetation structural outcome is possible when considering ecological theory and site-specific conditions carefully. For instance, a heavy seed supply into an abandoned farmland site might lead to structural development similar to managed stands (Sten_M/O in this case), at least for the first 78 years. We believe that site managers should make use of these basic rules in forest ecology when designing conservation projects.

In general, passive regeneration on Vorsø is progressing at a slow pace compared to other studies (Cramer, Hobbs & Standish, 2008; Verburg & Overmars, 2009). After 28 and 78 years, respectively, three spontaneous regenerated zones of Vorsø still hold shrub-dominated low vegetation and open areas. The timespan of 20–60 years for the development of stands of mid-successional tree species on abandoned farm land, concluded in previous studies (Cramer, Hobbs & Standish, 2008; Verburg & Overmars, 2009), fail to comply with most spontaneous regenerated areas on Vorsø, which are too dominated by gaps and low and sparse vegetation to fulfill this criteria. Successional arresting factors such as (high water level, drought and competition from herbs and grasses) are all present on Vorsø; however, as former farmland it should be categorized as having moderate conditions for colonization of woody vegetation, for example, compared to abandoned mining sites (Macdonald et al., 2015) or other areas exposed to heavy disturbances. In this light, we suggest a prolongation of the recognized timespan of natural afforestation under moderate conditions, such that the theory can comply with moderate conditions with a low degree of seed input from trees.

For biodiversity, the slow development might be beneficial, especially if early successional stages are underrepresented in a landscape level (Angelstam, 1998; Brawn, Robinson & Thompson, 2001). This is also supported by a study reviewing early-successional forest ecosystems, arguing that the prevailing focus on recovery of the closed-canopy stage should be shifted to greater focus on the qualities of the early-successional stages (Swanson et al., 2011). Besides, naturally developed gaps are found more valuable to biodiversity than artificial gaps created by thinning operations (Seidel, Ehbrecht & Puettmann, 2016). Thus, the spontaneous regeneration on Vorsø is generally in line with desired vegetation development according to international research, and of presumable interest to other afforestation projects in abandoned farmland sites in the temperate zone.

Rewilding of existing forests

This study evaluates rewilding of existing forests by comparing sites of unmanaged forest stands, which has previously been managed, to one stand (Sten_M/O) still under silvicultural management in terms of tree harvest, drainage and pest control. The results indicate a relatively high degree of vegetation structure heterogeneity in the V_O1 rewilded Vorsø forest. The stand is dominated by mature oak and ash trees, and shows the lowest median value on the axis 3 of PCA_10m (Fig. 3C), meaning that it has the most open canopy layer among the older stands, following disturbances by cormorant colonies, windfall and Dutch elm disease (Halberg & Gregersen, 2010). The reason for greater windfall impact here, for example, compared to the nearby managed Sten_M/O forest, is the increased water level, caused by drainage destruction, which prevents the roots from growing deep into the ground and causing existing roots to rot (Halberg & Gregersen, 2010). When we combine our results to the site history, our findings are in line with a Canadian study reporting that stands which have undergone canopy disturbance can evolve a high degree of spatial pattering (Zenner, 2005). According to international research, the impact of disturbances to biodiversity through unique forest structure is well documented (Berg et al., 1994; Navarro & Pereira, 2012; Nilsson, Hedin & Niklasson, 2001). We argue that the slow, moderate and persistent dynamics that have occurred in this rewilded stand (V_O1) have produced such vegetative attributes crucial to biodiversity (Fig. S2), and interpret it as being in line with the mediate disturbance hypothesis.

In contrast, V_O2 was subject to heavy and devastating disturbance from a packed colony of great cormorants in the 1980s, which entirely terminated the vegetation of tall-stemmed trees (Halberg & Gregersen, 2010). Instead of creating a unique structural composition, it rather pushed back the succession to the shrub or early stand stage (Fig. 2E). Our analyses cannot detect increased heterogeneity in this rewilded forest compared to a managed forest, and are therefore contrary to the findings in a review of post-disturbance vegetation structure, which state that regenerated vegetation is more heterogeneous than typical closed-canopy forest (Swanson et al., 2011). The divergence to our findings could be due to more homogenous stand replacement following the heavy impact of cormorant colonies compared to the regenerated vegetation following, for example, wildfires or windstorms.

Evaluation of the presented method

The capability of distinguishing forest areas by this method seems to be very sensitive, as it finds the overall similar vegetation development of the two young spontaneously regenerated zones (V_Y1 and V_Y2) to be different on six out of 10 parameters. The test of PCA_30m axes does not find any significant differences between these two former fields. Previous research shows that a scale of minimum 500 m2 is necessary to adequately detect defining structural attributes of different forest categories (Zenner, 2005). This is in line with our findings, where the 900-m2 grid cells confirm the similarity of the young Vorsø stands better than the 100-m2 grid cells, although the methods are different.

The few significant tests of V_O3 (0.38 ha; four cells in PCA_30m) and Fløj_Y4 (0.92 ha; 20 cells in PCA_30m) reveal that we reached the lower threshold of cell numbers. The 36 cells of V_O3 in PCA_10m apparently satisfy the statistical analysis to achieve trustworthy results. Increased point density of point clouds allows researchers to reduce cell size to one-m2 or even less (Müller & Brandl, 2009), providing 100 cells per 100 m2 instead of only one, as in this paper, though still in consideration of the above discussed scale issue. However, the high degree of significant results in the pairwise tests in combination with corresponding point densities in the literature, convince us of the sufficient point density in this study (in general between 0.5 and 0.9 per m2). A study from Colorado, USA, (Hall et al., 2005) calculated useful estimates of stand height, total above-ground biomass, foliage biomass and basal area from an average point density of 1.23 returns per m2. A study from the UK (Boyd & Hill, 2007) analyzed lidar intensity values captured over an area of woodland based on an average point cloud density of one return per 4.83 m2, and an Italian study (Mura et al., 2015) estimated indices of structural diversity by using an average density of 1.5 pulses per m2.

The PCA assessment ensures that the information in the point cloud retrieved by the calculated metrics is analyzed by non-correlated principal components, hereby avoiding that the metrics contribute with correlated information. Assumable, single lidar-derived metrics could explain the main part of the information instead of PCA axes. However, the wide number of metrics contributing to the eigenvectors in this study (Tables S5 and S6) also clarifies that the PCA axes express a complex interaction of metrics, which could be particular useful when dealing with low-density point clouds. A previous investigation succeeded in using PCA for choosing the three most significant metrics among nine metrics for modelling above-ground biomass across three forest study sites (Li, Andersen & McGaughey, 2008). We reason that the PCA axes are better suited for pairwise comparisons of forest sites than specific metrics, and that it is a different task to model concrete forest measurements, e.g., above-ground biomass, compared to designating key structural attributes suitable for differentiation of vegetation structure between stands. This is supported by the fact that two stands holding similar biomass can have very different structural composition. More research is needed to explore which methods are best suited for evaluating vegetation structural quality of forest sites using wall-to-wall lidar data.

As a disadvantage of the suggested simple method, we should mention the difficulties of ecological meaningful axes interpretation in some cases. In addition, it is a drawback that the PCA axes values cannot be directly compared to equivalent studies of other sites. The creation of a multi-site likewise analysis could form the foundation of a common reference dataset of PCA axes values, at least for delimited homogenous areas, for example, the northern temperate broadleaved tree species dominated zone. However, this is beyond the scope of this study.

Conclusion

The presented lidar-based evaluation of vegetation structure resulting from different afforestation strategies and management actions, successfully performs statistical tests of pairwise structural similarities and differences between forest stands; it points out key structural attributes and, at least to some extent, provides interpretable axes. The study takes advantage of already provided ground and site historical data, which turns out to constitute a well-functioning combination with the lidar analysis that can presumably be applied on a large number of sites world wide.

We find that spontaneous processes for establishment of forests in general leads to increased structural heterogeneity compared to forests under silvicultural management, as well as rewilding of existing forests leads to increased disturbance dynamics, which can result in increased vegetation heterogeneity or terminate and homogenize the vegetation.

Despite the large number of factors influencing spontaneous afforestation, this study indicates that underlying ecological factors controlling vegetation development might be predictable to a large degree. We therefore agree on the need to put more effort into evaluation of existing and future spontaneous regeneration projects in order to exploit and learn from these ecological field based laboratories (Cramer, Hobbs & Standish, 2008; Hobbs & Cramer, 2007b), hereby supporting managers in improving general conditions for biodiversity in areas under afforestation.

Supplemental Information

Supplemental Information 1 Plot of the four youngest forests and zones according to axes 2 and 5 of PCA_10m.

Click here for additional data file.

Supplemental Information 2 Maps of selected gradients expressed by axes of PCA_10m and PCA_30m.

Click here for additional data file.

Supplemental Information 3 Details on the 25 lidar-derived metrics used in this paper.

Click here for additional data file.

Supplemental Information 4 Results from pairwise tests of medians and variance of PCA_10m values between all forests and zones.

Click here for additional data file.

Supplemental Information 5 Results from pairwise tests of medians and variance of PCA_30m values between all forests and zones.

Click here for additional data file.

Supplemental Information 6 Interpretation of axes.

Click here for additional data file.

Supplemental Information 7 Eigenvectors of the first five axes of PCA_10m.

Click here for additional data file.

Supplemental Information 8 Eigenvectors of the first five axes of PCA_30m.

Click here for additional data file.

Supplemental Information 9 Raw lidar data filenames and source homepage.

Click here for additional data file.

Supplemental Information 10 Shape files for the site boundaries used in this study.

Click here for additional data file.

Thanks to Brody Steven Sandel for kindly helping with the R-statistical software and to Jens Gregersen for a thorough guidance on Vorsø. The municipality of Aarhus has kindly provided information about the Tranbjerg and Fløjstrup Forests. Thanks to Ari Arnold for linguistic review. Contains material from the Danish Department of Data Supply and Efficiency “Styrelsen for Dataforsyning og Effektivisering,” DHM-2007/Punktsky.

Additional Information and Declarations

Competing Interests

Author Contributions

Data Availability

The authors declare that they have no competing interests.

Henrik Thers conceived and designed the experiments, performed the experiments, analyzed the data, contributed reagents/materials/analysis tools, prepared figures and/or tables, authored or reviewed drafts of the paper, approved the final draft.

Peder Klith Bøcher conceived and designed the experiments, contributed reagents/materials/analysis tools, authored or reviewed drafts of the paper, approved the final draft.

Jens-Christian Svenning conceived and designed the experiments, authored or reviewed drafts of the paper, approved the final draft.

The following information was supplied regarding data availability:

The raw lidar data is freely available at: https://download.kortforsyningen.dk/content/dhm-2007punktsky.

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
