# Peer review of "Using lidar to assess the development of structural diversity in forests undergoing passive rewilding in temperate Northern Europe"

_PeerJ, doi:10.7717/peerj.6219_

## Round 0.1 · original submission · Major Revisions

I now receive 2 reviews and both of them suggest "Major Revisions", which I also concur. Please go over the comments and comply or rebut them accordingly.

Reviewer 1 ·

Basic reporting

Clear and unambiguous, professional English used throughout.

Experimental design

Here is where it comes my main concern to the methods side of this manuscript. Are you using all 2 grid cells data (10 x 10-meter n=7319 and 30 x 30-meter n=1002) be used in Principal Component Analysis (PCA)? You have a data set where you have to build the model now. In the future, you will get new samples that you have to predict using that prebuilt model. PCA is an unsupervised method although you may split data into the Training set and Test set to be sure it will generalize. It was no considered in MS.

Validity of the findings

The conclusions should be appropriately stated, should be connected to the original question investigated, and should be limited to those supported by the results.

Additional comments

This manuscript titled “Using lidar to assess the development of structural diversity in forests undergoing passive rewilding in temperate Northern Europe” combined LiDAR and Principal Component Analysis (PCA) to address structural diversity of forests. I find the topic of the paper very interesting and potentially useful by different users. Forest management is increasingly important for many purposes such as climate adaptation, modify stand structure and maintains biodiversity etc.

In this sense, providing a monitoring and interpreting structural development method for forest structural homogeneity and heterogeneity is very much welcome. This study is generally well written. However, I believe that the MS needs some clarifications and improvements before being considered for publication.

The main points to be addressed are:
Page 15- Lines 248-249. Stensballe area is no mention stand age range. In your manuscript, Stensballe was classed both into Mid-aged and Old stand (Table 1). In Figure 3., Stensballe is not consistent with your groups. The PCA outcomes did not indicated a good classifying for the Stensballe. You divided Vorsø into seven areas according to the history of the island but the development of structural diversity may be not consistent with your groups. In this sense my question is which is near the Stensballe real structure status ?? (management group or PCA outcome?)

Page 16- Lines 278-286. Here is where it comes my main concern to the methods side of this manuscript. Are you using all 2 grid cells data (10 x 10-meter n=7319 and 30 x 30-meter n=1002) be used in Principal Component Analysis (PCA)? You have a data set where you have to build the model now. In the future, you will get new samples that you have to predict using that prebuilt model. PCA is an unsupervised method although you may split data into the Training set and Test set to be sure it will generalize. It was no considered in MS.

Page 26- Lines 512-516. The eigenvectors of the PC3 of PCA_10m (Table S5.) shows the Percentile05_ExclGround and Percentile10_ExclGround loading greater than CanopyCover. It may not say PC3 of PCA_10m meaning that it has the most open canopy layer.

Table S5. The eigenvectors of the first five axes of PCA_10m show the PC1 have the highest proportion of variance (about 0.654). The PC2-PC5 are relative vary low proportion of variance (about 0.088-0.032). For assessing the development of structural diversity, the outcome may not be sufficiently.

Maybe you can try count pixels value (proportion) at different PCA axis and plot map. To show structural diversity consist of different PCA axes values distribution in forests.

Additionally I’ve found some minor concerns:
Figure 1, 4,5 is can't easy to read. The site location labels should be consistent with Figure 1.

Figure 3. The x-axes labels should be consistent with Figure 1.

Reviewer 2 ·

Basic reporting

The authors used air-borne lidar point cloud-derived metrics treated by PCA (principal component analysis) to analyze a unique Danish rewilding site, and try to find the relationship between the data and the existing vegetation development. The method is very straightforward but the manuscript is hard to read and understand.

Experimental design

The field experimental design is fine.

It’s difficult to understand the methods based on the description. I suggest the authors provide a figure to visualize the grid sampling method from line 254 to 268.

From line 261 to 264, it said "The 25 metrics were found in existing lidar literature and selected on a criteria of being used in comparison to manual ground surveys in previous studies of for being alike such metrics (Fig. S1)." However, I didn’t find what the "25 metrics" are in Figure S1. I didn’t find what the “25 metrics” in other part of this manuscript, neither.

Validity of the findings

It’s very hard to see which field site is young or old in the figures and tables. For example, for young stands 1/2/3, the abbreviation Y1/2/3 can be easier for readers to identify in figures and table.

Additional comments

I suggest the authors to resubmit the manuscript after major revision.

---

## Round 0.2 · Minor Revisions

I returned your first revision to both original reviewers and only received comments from one of them. Ideally, I wanted to wait for the comments from the other reviewer but they have confirmed that they are unable to re-review.

Anyway, the comments are positive but the reviewer still has doubts about the statistical analysis. Therefore, a "minor revision" is given.

Reviewer 1 ·

Basic reporting

Clear and unambiguous, professional English used throughout.

Experimental design

The submission should clearly define the research question, which must be relevant and meaningful.

Validity of the findings

The data analysis should be robust, statistically sound, and controlled.

Additional comments

1.I thank you for your responses, as you say “this study is interdisciplinary, meaning that it concerns forest ecology and remote sensing aspects at the same time”. Your results are compelling, the effects of different management strategy should be discussed the details in the manuscript. ie. The managed Sten_M/O forest stage is not the same as others due to tree harvest.

2.It’s difficult to understand the explains for relative low proportion of PCA axis (like as PC2-PC5) . The explanation is pretty thin. The Values of PCA_10m axis 5, which is a gradient from bare ground to low and sparse vegetation, subscribing intermediate values to all kinds of taller vegetation. LiDAR data are just in this way. For assessing the development of structural diversity, the outcome may not be sufficiently.

3.The manuscript is written in standard, unambiguous language. If there is a weakness, it is in the statistical analysis (as I have noted above) which should be improved upon before Acceptance.

---

## Round 0.3 · accepted · Accept

I have read over the manuscript and feel that the revision is ready for publication in PeerJ.

#